# Synergistic Combination of Irinotecan and Rapamycin Orally Delivered by Nanoemulsion for Enhancing Therapeutic Efficacy of Pancreatic Cancer

**DOI:** 10.3390/pharmaceutics15020473

**Published:** 2023-01-31

**Authors:** Yu-Hsuan Liu, Ling-Chun Chen, Wen-Ting Cheng, Pu-Sheng Wei, Chien-Ming Hsieh, Ming-Thau Sheu, Shyr-Yi Lin, Hsiu-O Ho, Hong-Liang Lin

**Affiliations:** 1School of Pharmacy, College of Pharmacy, Taipei Medical University, Taipei 11031, Taiwan; 2Department of Biotechnology and Pharmaceutical Technology, Yuanpei University of Medical Technology, Hsinchu 30015, Taiwan; 3Division of Gastroenterology, Department of Internal Medicine, Wan Fang Hospital, Taipei Medical University, Taipei 11696, Taiwan; 4Department of General Medicine, School of Medicine, College of Medicine, Taipei Medical University, Taipei 11031, Taiwan; 5School of Pharmacy, College of Pharmacy, Kaohsiung Medical University, Kaohsiung 80708, Taiwan

**Keywords:** irinotecan, rapamycin, silymarin, combination therapy, self-nanoemulsifying nanoemulsion, oral nano pharmaceuticals

## Abstract

In recent years, combining different types of therapy has emerged as an advanced strategy for cancer treatment. In these combination therapies, oral delivery of anticancer drugs is more convenient and compliant. This study developed an irinotecan/rapamycin-loaded oral lecithin-based self-nanoemulsifying nanoemulsion preconcentrate (_LB_SNENP_ir/ra_) and evaluated its synergistic combination effects on pancreatic cancer. _LB_SNENP loaded with irinotecan and rapamycin at a ratio of 1:1 (_LB_SNENP_ir10/ra10_) had a better drug release profile and smaller particle size (<200 nm) than the drug powder. Moreover, _LB_SNENP_ir10/ra10_ exhibited a strong synergistic effect (combination index [CI] < 1.0) in cell viability and combination effect studies. In the tumor inhibition study, the antitumor activity of _LB_SNENP_ir10/ra10/sily20_ against MIA PaCa-2 (a human pancreatic cancer cell line) was significantly increased compared with the other groups. When administered with rapamycin and silymarin, the area under the curve and the maximum concentration of irinotecan significantly improved compared with the control. We successfully developed an irinotecan/rapamycin-loaded oral self-nanoemulsifying nanoemulsion system to achieve treatment efficacy for pancreatic cancer.

## 1. Introduction

Pancreatic adenocarcinoma (PAC) is a highly fatal malignancy with a five-year overall survival rate of 9% irrespective of the disease stage [1,2]. PAC is the fourth leading cause of cancer-related death in the United States, resulting in an estimated 45,750 deaths each year [2]. Although surgery is the primary treatment option for long-term survival, less than 20% of patients with PAC qualify for initial resection at diagnosis [3]. For patients with unresectable PAC, especially metastatic PAC, chemotherapy is essential for prolonging life expectancy. However, first-line systemic chemotherapy with 5-fluorouracil- or gemcitabine-based regimens slightly prolongs the overall survival of patients with metastatic PAC. The propensity of PAC to develop chemoresistance and the highly malignant behavior of PAC have substantially reduced treatment effectiveness for this disease [4].

Concurrent treatment has been taken as the key measure of treating cancer due to its primary advantages of maximizing the efficacy and minimizing the toxicity at an adequate ratio. Combination chemotherapy regimens involving leucovorin, fluorouracil, irinotecan, oxaliplatin (FOLFIRINOX), and gemcitabine (GEM) plus nab-paclitaxel or erlotinib (Tarceva), have been demonstrated to improve the outcomes of patients with PAC [4,5,6]. Thus, FOLFIRINOX is among the primary standard treatments for unresectable PAC, and GEM plus nab-paclitaxel or erlotinib is another standard treatment option for PAC. Although combination therapies show better survival rate and quality of life (QoL), the overall improvement remains limited for advanced disease. Thus, effective strategies, new single agents, or new combination therapies are urgently required to markedly improve the clinical outcomes of patients with PACs.

The mammalian target of rapamycin (mTOR) signaling acts as the major regulator for cell proliferation. In the study, the patients with PAC and relatively high active phosphorylated mTOR^S2448^ showed significantly shorter survival and account for 15–20% [7]. Therefore, targeted mTOR treatment may have positive impact on clinical outcomes from patients carrying PAC. The mTOR-dependent signals stimulated hypoxia-inducible factor-1 (HIF-1)α accumulation and HIF-1-mediated transcription in cells with hypoxic conditions [8]. HIF-1 acts as the main regulator for inducing vascular endothelial growth factor in a hypoxia situation [9]. The majority characterization of PACs is surrounded by a hypoxic tumor microenvironment [10,11]. Hypoxic tumors are more aggressive than oxygenated tumors [12,13,14]. Suppression of the mTOR-HIF-1 signaling pathway might therefore be an effective therapeutic strategy for PAC.

Irinotecan exhibits antiangiogenic properties. The mTOR inhibitor, rapamycin, targeted the mTOR–HIF-1α axis and induced colon cancer cells to become sensitive to irinotecan, both in vitro and in vivo study using a xenografted metastasis model of human colorectal cancer. Moreover, low dose combinations made a significant portion of the tumor shrink, even in specimens resistant to irinotecan alone [15]. In a cytotoxicity study, the combination of irinotecan and rapamycin exerted a stronger killing effect on PSN1 cells between 48 and 72 h after incubation compared with irinotecan or rapamycin alone [16]. Jannier et al. reported that targeting the central node of the mTOR–HIF-1 axis with rapamycin and irinotecan suppressed the proliferation and metabolism of tumor cells [17]. Furthermore, Suzuki et al. demonstrated that the suppression of mTOR–HIF-1 signaling mediated the antitumor activity of metformin for PAC, revealing the different underlying mechanism from that of GEM. These results imply that the combination of rapamycin and irinotecan might be an effective therapeutic drug for PAC [18].

Currently, irinotecan is mainly administered through an intravenous bolus injection. Despite the advantages of oral chemotherapy over intravenous dosing [19,20], poor solubility and limited GI absorption can make oral therapy challenging. In the lumen of the gastrointestinal tract, oral bioavailability is often limited by the expression of ABC efflux transporters, such as P-glycoprotein (P-gp) and other metabolizing enzymes, including cytochrome P450 3A (CYP3A). Other limiting factors include the secretion from liver into bile via P-gp (ABCB1), the ATP-binding cassette drug-transporter C2 (ABCC2) and ATP-binding cassette drug-transporter G2 (ABCG2), and the first-pass effect in the liver [21,22,23,24]. In a previous study [24], the oral delivery of irinotecan was loaded in a self-emulsifying drug delivery system (SMEDDS) to enhance its solubility in combination with a P-gp/CYP3A dual-function inhibitor which was used to overcome the first-pass effect and increase the formation of the active metabolite SN-38. This method enhanced the antitumor effect due to the improved oral bioavailability of irinotecan, which made the active metabolite SN-38 form and accumulate.

In previous studies, a lecithin-based self-nanoemulsifying nanoemulsion preconcentrate (_LB_SNENP) was developed for the simultaneous loading of irinotecan and rapamycin with silymarin. This was used as the dual-function inhibitor for the oral delivery of the resultant self-nanoemulsifying nanoemulsion (_LB_SNENA) as a means of enhancing oral bioavailability [25,26,27,28]. We used this system and optimized it in the present study. Furthermore, most of the SMEDDSs used in this study, such as _LB_SNENA, are thermodynamically stable liquid formulations which demonstrated the high solubilization capacity of poorly soluble drugs. Thus, they can be directly filled into soft or hard gelatin capsules and act as a convenient method of oral administration.

## 2. Materials and Methods

### 2.1. Materials

Silymarin (80%) was purchased from Sanjaing (Jiaxing, China). Irinotecan hydrochloride and SN-38 were purchased from Scino Pharm (Tainan, Taiwan). SN38G was purchased from Cayman Chemical (Ann Arbor, MI, USA). Rapamycin was purchased from Chunghwa Chemical Synthesis and Biotech (New Taipei City, Taiwan). Capryol-90 was purchased from Gattefosse (Lyon, France). Tween 80 and camptothecin were purchased from Merck KGaA (Darmstadt, Germany). Cremophor EL was procured from Wei Ming Pharmaceutical (Taipei, Taiwan). Ascomycin was purchased from MedChemExpress (South Brunswick, NJ, USA). Soybean lecithin (Lipoid S-100) was purchased from Lipoid GmbH (Ludwigshafen, Germany). Dulbecco’s modified Eagle’s medium, fetal bovine serum, and horse serum were purchased from Corning (New York, NY, USA). Reagents used for high-performance liquid chromatography (HPLC) or ultra-performance liquid chromatography with tandem mass spectrometry (UPLC/MS/MS) were of HPLC or MS grade, and other reagents were of analytical grade.

### 2.2. Preparation of Irinotecan/Rapamycin-Loaded _LB_SNENP

Based on the results of previous studies, a mixture of lecithin and Tween 80 with cremophor EL as the surfactant system (SAA), and capryol-90 was selected as the oil phase, and propylene glycol (PG) as the cosurfactant [23]. Irinotecan/rapamycin-loaded oral _LB_SNENP was prepared using 18% capryol-90, 58% SAA, and 24% w/w PG with the drugs. These two drugs simultaneously dissolved in _LB_SNENP and formed an opalescent/translucent nanoemulsion. The irinotecan or rapamycin content was 10 mg in 1 g of _LB_SNENP. The mixture was heated in a 50–60 °C water bath until it completely dissolved into a light yellow and clear solution. Table 1 revealed the composition of 1 g of _LB_SNENA_bk_, _LB_SNENA_ir10_, and _LB_SNENA_ra10_.

### 2.3. Characterization of Irinotecan/Rapamycin-Loaded _LB_SNENA

In this study, 100 μL of _LB_SNENP_bk_, _LB_SNENP_ir10_, and _LB_SNENP_ra10_ was separately added to a 20-mL sample bottle containing 10 mL of double-distilled water. Then, the solution was gently shaken to obtain _LB_SNENA_bk_, _LB_SNENA_ir10_, and _LB_SNENA_ra10_, respectively. The average droplet size and size distribution of each formulation were measured at 25 °C by using an N5 submicron particle size analyzer (Beckman Coulter, Brea, CA, USA) at a scattering angle of 90°, and the intensity autocorrelation of the sample ranged from 5 × 10^4^ to 1 × 10^6^. Measurements were conducted three times for all the formulations to calculate the average diameter (nm), polydispersity index (PDI) and zeta potential (mV). The stability of _LB_SNENA_bk_, _LB_SNENA_ir10_, and _LB_SNENA_ra10_ was evaluated at room temperature for 1 month. At each time point, some samples were collected to determine the droplet size and the contents of irinotecan and rapamycin.

### 2.4. HPLC Instrumentation and Chromatographic Conditions

The contents of irinotecan and rapamycin in _LB_SNENP_ir_ and _LB_SNENP_ra_ were detected through HPLC. To determine the irinotecan content, we used the Waters 600E HPLC system with the SunFire C18 column (4.6 mm × 250 mm I.D., 5 μm; Waters). The mobile phase consisted of 10 mM phosphate buffer (pH 3.0)/acetonitrile/tetrahydrofuran (65/35/2 *v*/*v*), the flow rate was 0.8 mL/min, the column was maintained at 40 °C, the injection volume was 20 µL, and the fluorescence detector was set at the excitation and emission wavelengths of 370 and 470 nm, respectively. To determine the rapamycin content, we used the JASCO HPLC system with the Inersil ODS-2 C18 column (4.0 mm × 150 mm I.D., 5 μm). The mobile phase was a mixture of acetonitrile and water (20/80 *v*/*v*), the flow rate was 1.0 mL/min, the column was maintained at 55 °C, the injection volume was 100 µL, and the UV detector was set at the wavelength of 278 nm. Each data point came from the mean of at least three individual trials. The assay method was also well validated before. Analytical graphs are provided in the Appendix A.

### 2.5. Simultaneous Analysis of Irinotecan, SN38, SN38 Glucuronide, and Rapamycin in the Biosample through Ultra-Performance Liquid Chromatography with Tandem Mass Spectrometry

Chromatography was performed using a Waters Xevo TQ-XS with an Acquity UPLC system. Separation was performed using an Acquity UPLC BEH C18 column (2.1 × 100 mm I.D., 1.7 µm; Waters). The column was maintained at 55 °C, and the autosampler was set at 4 °C. The injection volume was 2 µL. Mobile phase A was 10 mM ammonium acetate (pH 3.0), and mobile phase B was acetonitrile. The gradient conditions of mobile phase B were as follows: 10% of mobile phase B for the first 1 min at a flow rate of 0.4 mL/min; mobile phase B was then linearly increased to 70% for 2 min and maintained for 1 min, and increased to 100% in 0.1 min at a flow rate of 0.7 mL/min and maintained for 1.5 min; the system finally returned to the initial condition in 2 min. The multiple reaction monitoring (MRM) scan mode was used for the quantification of the analytes irinotecan, SN38, SN38 glucuronide (SN38G), and rapamycin as well as the internal standards, camptothecin and ascomycin. Table 2 lists the protonated parents of the MS2 fragment ion MRM transitions for quantitation. The MS/MS instrument was operated using electrospray ionization in the positive mode, and the optimized parameters were as follows: desolvation temperature, 500 °C; source temperature, 150 °C; capillary voltage, 3.0 kV; cone gas flow, 150 L/h; and desolvation gas flow, 1000 L/h. Analytical graphs are provided in the Appendix A.

### 2.6. In Vitro Release of Irinotecan and Rapamycin from _LB_SNENP

The in vitro release of irinotecan and rapamycin from _LB_SNENP was examined using the dissolution method on United States Pharmacopeia (USP) Apparatus 2 (VK7000, Vankel, UK). The release medium of irinotecan was 500 mL of buffer (pH 1.2). The release medium of rapamycin was 0.4% sodium lauryl sulfate solution. The temperature of the dissolution medium was maintained at 37 °C ± 0.5 °C. The stirring rates for irinotecan and rapamycin were 50 and 100 rpm, respectively. Briefly, 3-mL aliquots of the sample were withdrawn for the assay at predetermined time points (0, 5, 10, 15, 30, 45, and 60 min) and replaced with the identical volume of the fresh medium. The contents of irinotecan and rapamycin were determined through HPLC as described in the earlier text. We filled 0.1 g of the irinotecan and rapamycin (10 mg/g) from _LB_SNENP in number 0 empty hard capsules. Two control groups were the irinotecan and rapamycin powder. Each dissolution data point was the mean of at least three individual trials.

### 2.7. Cell Viability and Combination Effect Studies

To evaluate the cytotoxicity and combination effects of rapamycin and silymarin as well as free irinotecan, we performed the 3-(4,5-dimethylthiazol-2-yl)-2,5-diphenyl-2H-tetrazolium bromide (MTT) assay by using the MIA PaCa-2 pancreatic cancer cell line. Briefly, the cells were seeded in a 96-well plate at a density of 3 × 10^4^ cells/well and incubated at 37 °C with 5% CO_2_ for 24 h. After 24 h, the cells were treated with different concentrations of free irinotecan (0–300 µM), rapamycin (0–300 µM), SN38 (0–300 nM), and silymarin (0–1000 µM). To determine the combination effect, the cells were treated with different ratios of irinotecan/rapamycin and SN38/rapamycin (1/0.3, 1/0.5, 1/1, and 1/2) and irinotecan/silymarin and SN38/silymarin (1/1 and 1/2). All the cells were incubated with the drugs for 24–48 h. Then, the MTT reagent and dimethyl sulfoxide were added for the formation and the dissolution of purple formazan crystals. The absorbance of each well was measured at 550 nm on a Cytation 3 cell imaging multimode reader (BioTek, Winooski, VT, USA). The half-maximal inhibitory concentration (IC_50_) and combination index (CI) of the drugs were calculated using CompuSyn Sofware (Paramus, NJ, USA) through the Chou–Talalay method. In the Chou–Talalay method, the CI values of <0.9, >1.1, and 0.9–1.1 indicated synergistic, antagonistic, and additive effects, respectively. Each data point was the mean of at least six individual experiments.

### 2.8. In Vivo Pharmacokinetic Studies

All animal experiments followed the protocol approved by the Laboratory Animal Center of Taipei Medical University (Approval No: LAC-2017-0334) and were conducted in compliance with the Animal Welfare Act. Eight-week-old male Sprague Dawley rats were used to investigate the pharmacokinetic (PK) profiles of irinotecan, SN38, SN38G, and rapamycin after single-dose oral administration. In total, 40 rats were randomized into eight groups (n = 5 per group): irinotecan solution in water (Sol_ir10_), rapamycin solution in water (Sol_ra10_), _LB_SNENP_ir10_, _LB_SNENP_ra10_, _LB_SNENP_ir10/ra5_, _LB_SNENP_ir10/ra10_, _LB_SNENP_ir10/ra5/sily20_, and _LB_SNENP_ir10/ra10/sily20_. Table 3 lists the dosing from different groups. All blood samples from the jugular vein were collected into K_2_EDTA blood collection tubes at 0.0833, 0.5, 1, 2, 3, 4, 6, 8, 10, 12, and 24 h after oral administration and were then stored at 4 °C. Next, 300 µL of the blood samples were added to 1.5-mL microtubes and immediately centrifuged at 6000 rpm for 10 min at 4 °C to obtain plasma. The plasma samples were stored at −80 °C until UPLC/MS/MS. PK parameters were calculated through a noncompartmental analysis using WinNonlin software (Pharsight, Princeton, NJ, USA). The results are expressed as mean ± standard deviation (SD). Relative bioavailability (F_RB_) was calculated using the following equation:$$F_{RB}=\frac{AUC_{A}/dose_{A}}{AUC_{B}/dose_{B}}\times 100\%$$

### 2.9. Bioanalysis of the Blood Concentrations of Irinotecan, SN-38, SN38G, and Rapamycin

To extract irinotecan, SN-38, and SN38G from the plasma samples, 100 μL of the plasma sample was mixed with 200 µL of acetonitrile for 3 min by using a multitube vortexer to extract analytes. After 6000-rpm centrifugation for 10 min at 4 °C, 0.1 mL of the supernatant was transferred to another 1.5-mL microtube and stored at 4 °C (analyte A). Subsequently, to collect rapamycin, 200 μL of the extracted solution (methanol/0.1 M zinc sulfate solution = 7/3) was added to 100 µL of the blood sample and vortexed for 1 min. The mixture was centrifuged at 6000 rpm at 4 °C for 10 min, and 100 μL of the supernatant was mixed with analyte A and vortexed for 10 s (analyte B). Subsequently, 10 μL of camptothecin (1 µg/mL) and 10 μL of ascomycin (3 µg/mL) were added to analyte B and then diluted with the mobile phase and mixed thoroughly. The final sample solution was injected into the UPLC/MS/MS system for analysis.

### 2.10. Tumor Inhibition Studies

All animal experiments followed the protocol approved by the Laboratory Animal Center of Taipei Medical University (Approval No: LAC-2017-0334) and were performed following animal care guidelines. Five-week-old male nu/nu mice received a subcutaneous injection of 100 μL (containing 15 × 10^5^ cells) of a MIA PaCa-2 cell suspension in Matrigel into their back. These tumor-bearing mice with a tumor volume of approximately 100 mm^3^ were randomized into seven groups: one control group (saline) and six experimental groups, namely _LB_SNENP_ir10_, _LB_SNENP_ra10_, _LB_SNENP_ir10/ra5_, _LB_SNENP_ir10/ra10_, _LB_SNENP_ir10/ra5/sily20_, and _LB_SNENP_ir10/ra10/sily20_ (n = 3–5 per group). Table 4 lists the dosing from different groups. Each formulation was orally administered 4 times every 3 days. The tumor volumes and body weights of the mice were measured every 3 days after the administration of the formulations. The tumor volume was calculated using the formula 1/2 length × width^2^. The mice were sacrificed through CO_2_ inhalation, and the tumors were harvested and weighed on day 31. The tumor growth inhibition rate (TGI%) was calculated as follows: (W_c_ − W_t_)/W_c_, where W_t_ is the tumor weight of each formulation group, and W_c_ is the tumor weight of the control group [23].

### 2.11. Statistical Analysis

Data are expressed as the mean ± SD of each study. Significant differences among the samples were determined using one-way analysis of variance (ANOVA). *p* value of 0.05 indicated statistical significance.

## 3. Results and Discussion

### 3.1. Characterization of _LB_SNENA

The optimized _LB_SNENP, composed of capryol-90, SAA, and PG at a weight ratio of 18/58/24, was selected to encapsulate irinotecan and rapamycin to form _LB_SNENA. These two drugs completely dissolved in _LB_SNENP, forming an opalescent/translucent nanoemulsion. As presented in Table 5, the mean droplet size (nm) and PDI of _LB_SNENA_ir_ and _LB_SNENA_ra_ were 122.7 ± 1.84 (0.212 ± 0.011) nm and 120.8 ± 2.25 (0.224 ± 0.010) nm, respectively, which were similar to those of _LB_SNENA_bk_ (149.3 ± 2.48 [0.305 ± 0.043] nm). The zeta potential (mV) of _LB_SNENA_ir_ and _LB_SNENA_ra_ were −4.14 ± 0.24 mV and −8.20 ± 0.30 mV, respectively, which were similar to those of _LB_SNENA_bk_ (−7.32 ± 0.46 mV). The results indicated that the drugs loaded in _LB_SNENP did not affect the self-nanoemulsifying property. Furthermore, during a 30-day period, the stability of the homogenous nanoemulsions of _LB_SNENA_ir_ and _LB_SNENA_ra_ was maintained at room temperature without precipitation, aggregation, or delamination. As presented in Figure 1A, the concentrations of irinotecan and rapamycin loaded in the nanoemulsions on day 30 did not differ from those on the initial day (>90%).

### 3.2. In Vitro Release of Irinotecan and Rapamycin from _LB_SNENP

The release of irinotecan and rapamycin (10 mg/g) from _LB_SNENP was examined using the USP dissolution method, and the results are illustrated in Figure 1B. The cumulative release rates of irinotecan and rapamycin from _LB_SNENP rapidly reached approximately 85% and 90%, respectively, at 15 min. Both the formulations were completely released within 30 min. The dissolution percentages of raw irinotecan and rapamycin powders were only approximately 63% and 32%, respectively, at the endpoint. Compared with the raw powders, _LB_SNENP_ir_ and _LB_SNENP_ra_ exhibited a significant difference in their dissolution percentages (*p* < 0.05). The results indicate that _LB_SNENP can considerably improve the solubility and release rate of hydrophobic drugs.

### 3.3. Cell Viability and Combination Effect Studies

Cell viability and combination effects were determined using the MTT assay. We evaluated the antitumor effects of irinotecan, SN38, rapamycin, and silymarin at different ratios on a human pancreatic cancer cell line (MIA PaCa-2). SN38 is an active metabolite of irinotecan, and its activity is approximately 100–1000 times that of irinotecan. Therefore, we designed the sample concentration range of irinotecan, rapamycin, and silymarin in micromolarity and that of SN38 in nanomolarity. The IC50 values of irinotecan, SN38, rapamycin, and silymarin against MIA PaCa-2 cells were 4.78 ± 9.42, 51.71 ± 33.41, 2.98 ± 0.97, and 216.73 ± 32.78 μM, respectively, at 24 h and 12.20 ± 4.56, 50.74 ± 10.64, 3.88 ± 1.17, and 153.08 ± 34.13 μM, respectively, at 48 h.

To investigate combination effects, we mixed rapamycin with 300 μM irinotecan or 300 nM SN38 at different ratios (0.3:1, 0.5:1, 1:1, and 2:1). Subsequently, we combined silymarin with irinotecan or SN38 at the ratios of 1:1 and 1:2 and then serially diluted and co-delivered them to treat the cells. The results of rapamycin/irinotecan and rapamycin/SN38 treatments are presented in Table 6. The CI value of rapamycin/irinotecan ranged from 0.11 (concentration ratio = 0.3:1 at 24 h) to 0.28 (concentration ratio = 2:1 at 24 h). The CI value of rapamycin/SN38 at 24 h appeared to be similar to that of rapamycin/irinotecan. Except for irinotecan/silymarin at the ratio of 1:2, the CI value of silymarin combined with irinotecan or SN38 (1:1 or 1:2) was <1.0 at 24 h. After 48-h treatment, all the groups exhibited greater improvement in the inhibition of cell proliferation; thus, their CI values were <0.1. The results indicate that the aforementioned combinations are considerably effective against MIA PaCa-2 cells when _LB_SNENP is used as the co-delivery system for irinotecan, rapamycin, and silymarin.

### 3.4. In Vivo PK studies

The 8-week-old male Sprague Dawley rats with jugular vein catheters were used as the experimental animals. They were randomly divided into eight groups (n = five per group) and were administered the drugs once through oral gavage. The PK profiles and related PK parameters for irinotecan (Figure 2A and Table 7), SN38 (Figure 2B and Table 8), SN38G (Figure 2C and Table 9), and rapamycin (Figure 2D and Table 10) are illustrated in Figure 2 and listed in Table 3, Table 4, Table 5 and Table 6, respectively. According to the irinotecan injection package insert, plasma samples would mainly comprise the prototype of irinotecan, followed by SN38G and SN38. In our studies, the results of the distribution of irinotecan, SN38G, and SN38 in plasma measured through UPLC/MS/MS were compatible with the data provided in the injection package insert. Notably, the pharmacokinetics parameters calculated for irinotecan and rapamycin showed the comparable trend of the maximum serum concentration (C_max_), which demonstrated that _LB_SNENP developed in this study obviously increased the absorption of irinotecan and rapamycin, resulting in the increase of C_max_ by approximately 2.0–5.7 and 3–10 times, respectively, compared with that of irinotecan and rapamycin aqueous solution (Sol_ir10_ and Sol_ra10_, respectively). However, _LB_SNENP groups with rapamycin also exhibited the higher C_max_ values of irinotecan and SN38, compared with the _LB_SNENP group without rapamycin, which may be attributed to the combination of rapamycin. As previously reported, rapamycin inhibited the P-glycoprotein by competitive inhibition, which efficiently decreased the elimination of irinotecan [29]. Thus, increasing the amount of rapamycin evidently increased the area under the curve (AUC_0→∞_) of irinotecan and SN38. SN38, a major active metabolite of irinotecan, is metabolized by carboxylesterases. Irinotecan was metabolized by the enzymes encoded by the UGT1A1 and CYP3A4 genes to form the inactive metabolites SN38G, APC, and NPC. Therefore, the increase in AUC of SN38 could lead to the better anticancer effect. Based on the area under the curve (AUC_0→∞_) of Sol_ir10_, the relative bioavailability (*F_RB_*) of irinotecan, SN38, and rapamycin loaded in _LB_SNENP was calculated using the formula previously described. The *F_RB_* of irinotecan, SN38, and rapamycin was enhanced by approximately 2.0–5.0, 1.1–3.0, and 1.5–4.3 times, respectively. The conversion efficiency of SN38 is a key point in the efficiency of cancer therapy. The results of the conversion efficiency of SN38 are listed in Table 8. Irinotecan (10 mg/g), combined with rapamycin (10 mg/g) and silymarin (20 mg/g) loaded in _LB_SNENP, exhibited the highest value (75.2%), followed by _LB_SNENP_ir10/ra10_ (61.9%). Although the *F_RB_* of _LB_SNENP_ir10/ra10/sily20_ was lower than that of _LB_SNENP_ir10/ra10_, the higher conversion efficiency might enhance the blood concentration of SN38 to improve antitumor efficacy. Furthermore, SN38G is the inactive metabolite of SN38 [30]. The values of C_max_ and AUC of SN38G in the _LB_SNENP groups with rapamycin were higher than those in the _LB_SNENP group without rapamycin, which resulted from the higher amount of SN38. In particular, the time in which maximum plasma concentration (T_max_) was reached was delayed significantly in those _LB_SNENP groups with rapamycin, indicating that the addition of rapamycin delayed the metabolism of SN38. This result was consistent with the previous report, which showed that the mTOR inhibitor could also act as UGT1A1 inhibitor [31].

In the PK study of rapamycin administered in combination with irinotecan and silymarin, the C_max_ and AUC_0→∞_ of rapamycin showed gradual downward trends. These phenomena have been reported in the literature. When the P-gp inhibitor (rapamycin) was co-delivered with the chemotherapeutic reagent (irinotecan), it prevented the elimination of antitumor drugs and promoted the accumulation of drugs in the body. In conclusion, with the synergistic effects of P-gp and the mTOR inhibitor (rapamycin) and CYP3A4 inhibitor (silymarin), irinotecan loaded in _LB_SNENP increased the C_max_, AUC_0→∞_, *F_RB_*, and conversion efficiency of SN38; this might also improve the efficacy of combination therapy.

### 3.5. In Vivo Therapeutic Studies

The in vivo tumor inhibition studies of the multiple drugs loaded in _LB_SNENP were performed using a nude mouse model bearing MIA PaCa-2 xenografts. The tumor growth curves are illustrated in Figure 3A. After the oral administration of all the formulations was performed four times, except in the control group (saline), the treatment groups exhibited substantial inhibition in the growth of MIA PaCa-2 cells. The TGI% on day 12 for _LB_SNENP_ir10/ra10/sily20_ was 88.4%. The TGI% on day 12 was 84.3%, 80.0%, 71.9%, 82.8%, and 79.8% for _LB_SNENP_ir10/ra5/sily20_, _LB_SNENP_ir10_, _LB_SNENP_ra10_, _LB_SNENP_ir10/ra10_, and _LB_SNENP_ir10/ra5_, respectively. At the same dose of irinotecan combined with rapamycin, _LB_SNENP_ir10/ra10/sily20_ and _LB_SNENP_ir10/ra5/sily20_ exhibited greater antitumor activity than that of dual drugs (_LB_SNENP_ir10/ra10_ and _LB_SNENP_ir10/ra5_). During the regular observation period (every 3 days), the tumor growth rate was slower in the treatment group of _LB_SNENP_ir10/ra10/sily20_ than in the other groups after the last administration. On day 31, we sacrificed the mice through CO_2_ inhalation, and the tumors were harvested and weighed. Figure 3B illustrates the excised tumor mass. Significant differences in tumor weights were observed between the combination and control groups (*p* < 0.05). During the 30-day experimental period, changes were observed in the mouse weight across each experimental group. These are shown in Figure 3C. The treatment groups exhibited a slight weight loss of not more than 20%. The survival rate is presented in Figure 3D. One mouse in the _LB_SNENP_ir10/ra10/sily20_ group died on day 6, and one mouse in the _LB_SNENP_ir10/ra10_ group died on day 9. Diarrhea is a severe side effect of irinotecan. No diarrhea was noted in the nude mice in all the experimental groups. These results indicate that _LB_SNENP–encapsulated irinotecan, rapamycin, and silymarin not only reduced the dose of the drugs to achieve treatment efficacy but also considerably reduced side effects.

## 4. Conclusions

In this study, we successfully developed an oral lecithin-based self-nanoemulsifying nanoemulsion drug delivery system (_LB_SNENA), which could simultaneously encapsulate one or more hydrophobic drugs to improve their solubility and oral bioavailability. This drug delivery system had a nanoscale particle size and high stability. Furthermore, the in vivo and in vitro studies of irinotecan, rapamycin, and silymarin loaded in _LB_SNENP exhibited their significant anticancer synergistic effects on human pancreatic cancer cells. The combination of multiple drugs with different pharmacological mechanisms considerably increased the inhibition of tumor proliferation, reduced the dosage of a single drug, and prevented the occurrence of side effects. Thus, irinotecan combined with rapamycin and silymarin, loaded in _LB_SNENP to form a self-nanoemulsifying nanoemulsion, is a potential drug delivery system for the oral administration of chemotherapy drugs and exerts synergistic antitumor effects.

## Figures and Tables

**Figure 1 pharmaceutics-15-00473-f001:**
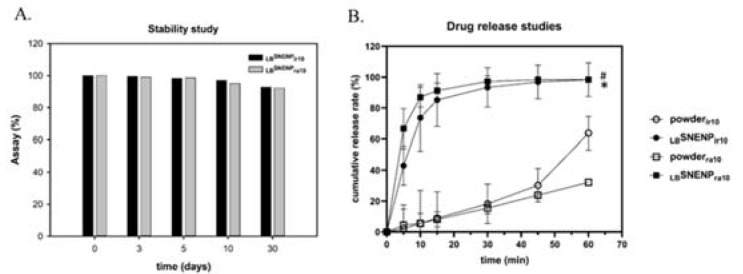
Content of irinotecan or rapamycin-loaded _LB_SNENP in the stability test (**A**); drug release profiles of the irinotecan powder, rapamycin powder, _LB_SNENP_ir_, and _LB_SNENP_ra_ (**B**). * *p* < 0.05 when _LB_SNENP_ir_ was compared with the irinotecan powder. # *p* < 0.05 when _LB_SNENP_ra_ was compared with the rapamycin powder. Each point is shown as mean ± standard deviation (n = 3). Abbreviations: _LB_SNENP, lecithin-based self-nanoemulsifying nanoemulsion preconcentrate; powder is pure drug; ir, irinotecan; ra, rapamycin; number is dose.

**Figure 2 pharmaceutics-15-00473-f002:**
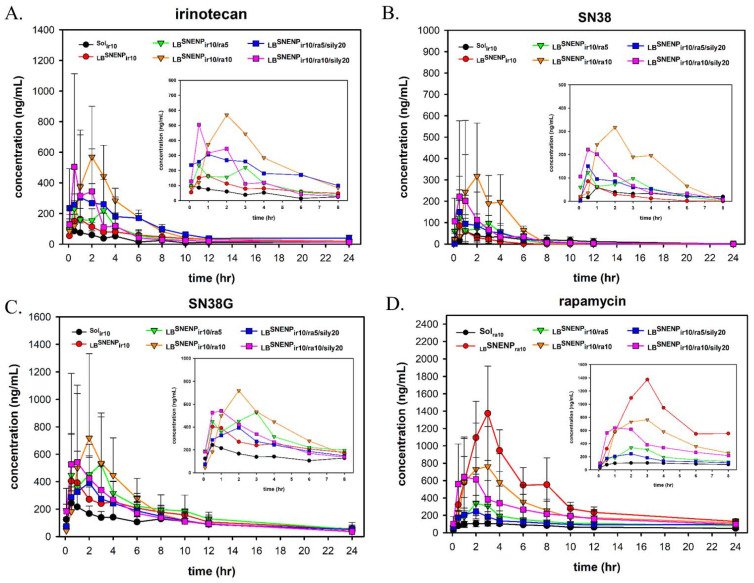
In vivo PK profiles of irinotecan (**A**), SN38 (**B**), SN38G (**C**), and rapamycin (**D**) after oral administration with Sol_ir10_, Sol_ra10_, _LB_SNENP_ir10_, _LB_SNENP_ra10_, _LB_SNENP_ir10/ra5_, _LB_SNENP_ir10/ra10_, _LB_SNENP_ir10/ra5/sily20_, and _LB_SNENP_ir10/ra10/sily20_. Each point is shown as mean ± standard deviation (n = 3–5). Abbreviations: Sol, solution; _LB_SNENP, lecithin-based self-nanoemulsifying nanoemulsion preconcentrate; ir, irinotecan; ra, rapamycin; sily, silymarin; number is dose.

**Figure 3 pharmaceutics-15-00473-f003:**
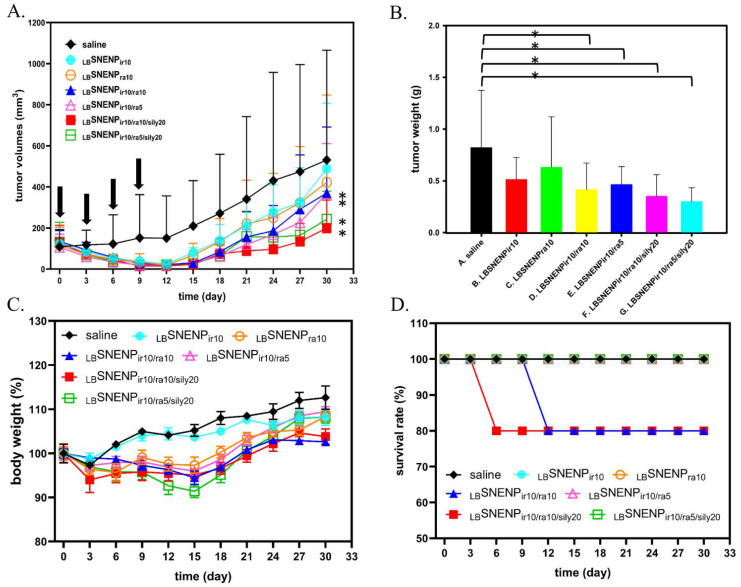
Tumor inhibition studies (**A**); tumor weight (**B**); body weight changes (**C**); and survival rate (**D**) of nude mice bearing Mia PaCa-2 tumor xenografts after oral administration with saline, _LB_SNENP_ir10_, _LB_SNENP_ra10_, _LB_SNENP_ir10/ra5_, _LB_SNENP_ir10/ra10_, _LB_SNENP_ir10/ra5/sily20_, and _LB_SNENP_ir10/ra10/sily20_ (on days 0, 3, 6, and 9). Each point is shown as mean ± standard deviation (n = 4 to 5). * *p* < 0.05 when _LB_SNENP_ir10/ra5_, _LB_SNENP_ir10/ra10_, _LB_SNENP_ir10/ra5/sily20_, and _LB_SNENP_ir10/ra10/sily20_ were compared with saline. Abbreviations: _LB_SNENP, lecithin-based self-nanoemulsifying nanoemulsion preconcentrate; ir, irinotecan; ra, rapamycin; sily, silymarin; number is dose.

**Table 1 pharmaceutics-15-00473-t001:** Composition of 1 g _LB_SNENA_bk_, _LB_SNENA_ir10_, and _LB_SNENA_ra10_.

Formulations	Drug		Oil (18%)	Co-Surfactant (24%)	Surfactants (58% SAA)		
Composition	Irinotecan	Rapamycin	Capryol 90	Propylene glycol	Lecithin	Tween 80	Cremophor EL
_LB_SNENA_bk_	-	-	0.182 g	0.242 g	0.198 g	0.286 g	0.092 g
_LB_SNENA_ir10_	10 mg	-	0.182 g	0.242 g	0.198 g	0.286 g	0.092 g
_LB_SNENA_ra10_	-	10 mg	0.182 g	0.242 g	0.198 g	0.286 g	0.092 g

Abbreviations: _LB_SNENA, lecithin-based self-nanoemulsifying nanoemulsion; bk, blank; ir, irinotecan; ra, rapamycin; number is dose.

**Table 2 pharmaceutics-15-00473-t002:** Optimized MRM parameters of irinotecan, SN38, SN38G, rapamycin, camptothecin, and ascomycin.

Compounds	Formula	Parent *m*/*z*	Daughters *m*/*z*	Cone Volt.	Col. Energy
Irinotecan	C_33_H_38_N_4_O_6_	587.31	124.10	76	34
SN38	C_22_H_20_N_2_O_5_	393.21	349.14	70	26
SN38G	C_28_H_28_N_2_O_11_	569.20	393.17	78	28
Rapamycin	C_51_H_79_NO_13_	931.60	864.50	35	16
Camptothecin	C_20_H_16_N_2_O_4_	349.14	305.15	52	22
Ascomycin	C_43_H_69_NO_12_	809.51	756.49	48	22

Note: the analytes are irinotecan, SN38, SN38 glucuronide (SN38G), and rapamycin. The internal standards are camptothecin and ascomycin.

**Table 3 pharmaceutics-15-00473-t003:** The dose of irinotecan, rapamycin, and silymarin in each group of PK studies.

Dose	1 g dd H_2_O		1 g _LB_SNENP					
	Sol_ir10_	Sol_ra10_	_LB_SNENP_ir10_	_LB_SNENP_ra10_	_LB_SNENP_ir10/ra5_	_LB_SNENP_ir10/ra10_	_LB_SNENP_ir10/ra5/sily20_	_LB_SNENP_ir10/ra10/sily20_
Irinotecan (mg/kg)	10	-	10	-	10	10	10	10
Rapamycin (mg/kg)	-	10	-	10	5	10	5	10
Silymarin (mg/kg)	-	-	-	-	-	-	20	20

Abbreviations: Sol, solution; _LB_SNENP, lecithin-based self-nanoemulsifying nanoemulsion preconcentrate; ir, irinotecan; ra, rapamycin; sily, silymarin; number is dose.

**Table 4 pharmaceutics-15-00473-t004:** The dose of irinotecan, rapamycin, and silymarin in each group of efficacy studies.

	Groups						
Dose	Saline	_LB_SNENP_ir10_	_LB_SNENP_ra10_,	_LB_SNENP_ir10/ra5_	_LB_SNENP_ir10/ra10_	_LB_SNENP_ir10/ra5/sily20_	_LB_SNENP_ir10/ra10/sily20_
Irinotecan (mg/kg)	-	10	-	10	10	10	10
Rapamycin (mg/kg)	-	-	10	5	10	5	10
Silymarin (mg/kg)	-	-	-	-	-	20	20

Abbreviations: _LB_SNENP, lecithin-based self-nanoemulsifying nanoemulsion preconcentrate; ir, irinotecan; ra, rapamycin; sily, silymarin; number is dose.

**Table 5 pharmaceutics-15-00473-t005:** The mean droplet size, PDI, and zeta potential of _LB_SNENA.

Sample	Droplet Size (nm)	PDI	Zeta Potential (mV)
_LB_SNENA_bk_	149.3 ± 2.48	0.305 ± 0.043	−7.32 ± 0.46
_LB_SNENA_ir10_	122.7 ± 1.84	0.212 ± 0.011	−4.14 ± 0.24
_LB_SNENA_ra10_	120.8 ± 2.25	0.224 ± 0.010	−8.20 ± 0.30

Abbreviations: _LB_SNENA, lecithin-based self-nanoemulsifying nanoemulsion; bk, blank; ir, irinotecan; ra, rapamycin; number is dose.

**Table 6 pharmaceutics-15-00473-t006:** Combination index (CI_50_) for different ratios of irinotecan, SN38, rapamycin, and silymarin at 24 and 48 h.

Drug	Ratio	CI_50_	
		24 h	48 h
rapamycin:irinotecan	0.3:1	0.11	0.02
	0.5:1	0.16	0.01
	1:1	0.10	0.07
	2:1	0.28	0.01
rapamycin:SN38	0.3:1	0.20	0.01
	0.5:1	0.21	0.04
	1:1	0.35	0.03
	2:1	0.23	0.04
irinotecan:silymarin	1:1	0.69	0.01
SN38:silymarin	2:1	1.44	0.00
	1:1	0.97	0.09
	2:1	0.84	0.14

**Table 7 pharmaceutics-15-00473-t007:** PK parameter estimations of irinotecan after oral administration with Sol_ir10_, _LB_SNENP_ir10_, _LB_SNENP_ir10/ra5_, _LB_SNENP_ir10/ra10_, _LB_SNENP_ir10/ra5/sily20_, and _LB_SNENP_ir10/ra10/sily20_.

	Sol	_LB_SNENP				
Title	ir10	ir10	ir10/ra5	ir10/ra10	ir10/ra5/sily20	ir10/ra10/sily20
C_max_ (ng/mL)	106.4 ± 86.6	205.3 ± 86.6	349.7 ± 257.8	606.2 ± 283.6	360.1 ± 104.0	576.5 ± 497.0
T_max_ (h)	1.2 ± 1.6	1.2 ± 0.8	2.1 ± 1.2	2.0 ± 1.0	2.0 ± 1.0	1.0 ± 0.9
AUC_0→∞_ (ng·h/mL)	553.2 ± 311.9	1097.6 ± 369.7	1250.7 ± 526.4	2747.1 ± 948.7	1962.6 ± 1137.9	1361.2 ± 744.6
t_1/2_ (h)	3.5 ± 1.1	2.8 ± 0.6	1.9 ± 0.3	2.1 ± 0.2	2.2 ± 1.1	2.2 ± 1.5
F_RB_ (%)	100.0%	198.4%	226.1%	496.6%	354.8%	246.1%

Each point is shown as mean ± standard deviation (n = 3–5). Abbreviations: Sol, solution; _LB_SNENP, lecithin-based self-nanoemulsifying nanoemulsion preconcentrate; ir, irinotecan; ra, rapamycin; sily, silymarin; number is dose; F_RB_, relative bioavailability.

**Table 8 pharmaceutics-15-00473-t008:** PK parameter estimations of SN38 after oral administration with Sol_ir10_, _LB_SNENP_ir10_, _LB_SNENP_ir10/ra5_, _LB_SNENP_ir10/ra10_, _LB_SNENP_ir10/ra5/sily20_, and _LB_SNENP_ir10/ra10/sily20_.

	Sol	_LB_SNENP				
Title	ir10	ir10	ir10/ra5	ir10/ra10	ir10/ra5/sily20	ir10/ra10/sily20
C_max_ (ng/mL)	21.6 ± 3.4	141.6 ± 112.0	193.4 ± 112.2	357.7 ± 284.3	161.5 ± 125.0	202.0 ± 308.6
T_max_ (h)	1.5 ± 0.7	0.8 ± 0.4	1.3 ± 1.4	1.5 ± 0.7	1.4 ± 0.8	1.0 ± 0.7
AUC_0→∞_ (ng·h/mL)	137.9 ± 99.1	209.4 ± 121.9	412.1 ± 220.2	1136.5 ± 815.9	502.6 ± 333.7	684.5 ± 553.1
t_1/2_ (h)	4.0 ± 2.1	1.9 ± 0.6	1.1 ± 0.2	1.1 ± 0.7	2.3 ± 1.6	1.2 ± 0.4
F_RB_ (%)	100.0%	151.8%	298.8%	824.1%	364.5%	496.4%
CE (%)	37.3%	28.5%	49.3%	61.9%	38.3%	75.2%

Each point is shown as mean ± standard deviation (n = 3–5). Abbreviations: Sol, solution; _LB_SNENP, lecithin-based self-nanoemulsifying nanoemulsion preconcentrate; ir, irinotecan; ra, rapamycin; sily, silymarin; number is dose; F_RB_, relative bioavailability; CE, conversion efficiency.

**Table 9 pharmaceutics-15-00473-t009:** PK parameter estimation of SN38G after oral administration with Sol_ir10_, _LB_SNENP_ir10_, _LB_SNENP_ir0/ra5_, _LB_SNENP_ir10/ra10_, _LB_SNENP_ir10/ra5/sily20_, and _LB_SNENP_ir10/ra10/sily20_.

	Sol	_LB_SNENP				
Title	ir10	ir10	ir10/ra5	ir10/ra10	ir10/ra5/sily20	ir10/ra10/sily20
C_max_ (ng/mL)	245.1 ± 117.1	454.5 ± 302.0	708.8 ± 318.5	985.7 ± 518.7	491.9 ± 204.0	789.2 ± 636.1
T_max_ (h)	0.4 ± 0.2	0.7 ± 0.4	2.3 ± 0.6	2.3 ± 1.0	2.3 ± 1.3	1.2 ± 0.8
AUC_0→∞_ (ng·h/mL)	2369.0 ± 652.8	3546.7 ± 1014.1	3729.3 ± 172.6	4359.9 ± 2547.8	3262.7 ± 1472.8	4058.6 ± 1718.4
t_1/2_ (h)	5.2 ± 1.7	7.0 ± 2.1	5.1 ± 0.8	3.0 ± 0.4	4.7 ± 1.2	3.7 ± 0.7

Each point is shown as mean ± standard deviation (n = 3–5). Abbreviations: Sol, solution; _LB_SNENP, lecithin-based self-nanoemulsifying nanoemulsion preconcentrate; ir, irinotecan; ra, rapamycin; sily, silymarin; number is dose.

**Table 10 pharmaceutics-15-00473-t010:** PK parameter estimations of rapamycin after oral administration with Sol_ra10_, _LB_SNENP_ra10_, _LB_SNENP_ir10/ra5_, _LB_SNENP_ir10/ra10_, _LB_SNENP_ir10/ra5/sily20_, and _LB_SNENP_ir10/ra10/sily20_.

	Sol	_LB_SNENP				
Title	ra10	ra10	ir10/ra5	ir10/ra10	ir10/ra5/sily20	ir10/ra10/sily20
C_max_ (ng/mL)	113.6 ± 26.4	1185.6 ± 604.2	747.4 ± 441.8	1107.1 ± 308.0	344.4 ± 37.6	1140.9 ± 863.7
T_max_ (h)	3.3 ± 2.3	2.5 ± 0.7	2.3 ± 0.6	2.3 ± 1.0	1.8 ± 0.4	1.5 ± 0.6
AUC_0→∞_ (ng·h/mL)	1689.5 ± 176.1	7304.0 ± 4835.6	4308.2 ± 1970.9	6598.2 ± 1092.2	2767.6 ± 700.6	6499.4 ± 2306.1
t_1/2_ (h)	6.1 ± 3.9	4.3 ± 0.8	6.6 ± 1.9	3.4 ± 1.2	7.7 ± 4.4	4.8 ± 3.2
F_RB_ (%)	100.0%	432.3%	255.0%	390.5%	163.8%	384.7%

Each point is shown as mean ± standard deviation (n = 3–5). Abbreviations: Sol, solution; _LB_SNENP, lecithin-based self-nanoemulsifying nanoemulsion preconcentrate; ir, irinotecan; ra, rapamycin; sily, silymarin; number is dose; F_RB_, relative bioavailability.

## Data Availability

Source data are available from the corresponding author upon reasonable request.

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
