# Peer review of "Synergistic Combination of Irinotecan and Rapamycin Orally Delivered by Nanoemulsion for Enhancing Therapeutic Efficacy of Pancreatic Cancer"

_pharmaceutics, 2023, doi:10.3390/pharmaceutics15020473_

Round 1

Reviewer 1 Report

The dissolution experiment is problematic. USP dissolution is a paddle method, not designed for dissolution of nanoemulsion. Cremophor EL is not appropriate for nanoemulsion formulation. The amount of drug in each batch is 10 mg/1 g total material. This drug content is too low to have any potential effect. The dissolution figure 1A legends need to be corrected. The authors simply collected 3 mL dissolution samples from the 500 mL total medium and analyzed. How can you confirm that the drug was in the solution form? Overall, the dissolution method is inappropriate to study drug releae from nanoemulsion.  

Reviewer 2 Report

Submission requires revision. PDF attached.

Round 2

Reviewer 2 Report

The authors carefully addressed the comments. Provided additional data and tables as requested. Hence, the manuscript is acceptable in its current form.